# Incidence of and trends in hip fracture among adults in urban China: A nationwide retrospective cohort study

Chenggui Zhang[1☯], Jingnan Feng[2☯], Shengfeng Wang[2☯], Pei Gao[2☯], Lu Xu[2], Junxiong Zhu[1], Jialin Jia[1], Lili Liu[2], Guozhen Liu[3], Jinxi Wang[4], Siyan Zhan[2,5‡*], Chunli Song[1‡*]

1 Department of Orthopedics, Peking University Third Hospital, Beijing, China, 2 Department of Epidemiology and Biostatistics, School of Public Health, Peking University, Beijing, China, 3 Peking University Health Information Technology, Beijing, China, 4 Beijing Healthcom Data Technology, Beijing, China, 5 Research Center of Clinical Epidemiology, Peking University Third Hospital, Beijing, China

☯ These authors contributed equally to this work.
‡ These authors are joint senior authors on this work.
* siyan-zhan@bjmu.edu.cn (SZ); schl@bjmu.edu.cn (CS)

## Abstract

### Background

Hip fracture is a public health concern because of its considerable morbidity, excess mortality, great risk of disability, and high societal healthcare costs. China has the largest population of older people in the world and is experiencing rapid population aging and facing great challenges from an increasing number of hip fractures. However, few studies reported the epidemiology, especially at a national level. We aimed to evaluate trends in hip fracture incidence and associated costs for hospitalization in China.

### Methods and findings

We conducted a population-based study using data between 2012 and 2016 from the national databases of Urban Employee Basic Medical Insurance and Urban Resident Basic Medical Insurance in China, covering about 480 million residents. Data from around 102.56 million participants aged 55 years and older during the study period were analyzed. A total of 190,560 incident hip fracture patients (mean age 77.05 years, standard deviation 8.94; 63.99% female) were identified. Primary outcomes included the age- and sex-specific incidences of hip fracture. Associated annual costs for hospitalization were also calculated. Incidence was described as per 100,000 person-years at risk, and 95% confidence intervals were computed assuming a Poisson distribution. Hip fracture incidence overall in China did not increase during the study period despite rapid population aging. Incidence per 100,000 was 180.72 (95% CI 137.16, 224.28; P < 0.001) in 2012 and 177.13 (95% CI 139.93, 214.33; P < 0.001) in 2016 for females, and 121.86 (95% CI 97.30, 146.42; P < 0.001) in 2012 and 99.15 (95% CI 81.31, 116.99; P < 0.001) in 2016 for males. For both sexes, declines in hip fracture incidence were observed in patients aged 65 years and older, although incidence was relatively stable in younger patients. However, the total absolute

**Data Availability Statement:** Summarized health data about hip fracture can be accessed by contacting the National Insurance Claims for Epidemiological Research (NICER) Group, School

of Public Health, Peking University. Contact email: 0016163159@bjmu.edu.cn.

**Funding:** SZ is funded by the China National Natural Science Foundation (91646107; http://www.nsfc.gov.cn). CS is funded by the China National Natural Science Foundation (81874010; http://www.nsfc.gov.cn). The funders had no role in study design, data collection and analysis, decision to publish, or preparation of the manuscript.

**Competing interests:** The authors have declared that no competing interests exist.

**Abbreviations:** UEBMI, Urban Employee Basic Medical Insurance; URBMI, Urban Resident Basic Medical Insurance.

number of hip fractures in those 55 years and older increased about 4-fold. The total costs for hospitalization showed a steep rise from US$60 million to US$380 million over the study period. Costs for hospitalization per patient increased about 1.59-fold, from US$4,300 in 2012 to US$6,840 in 2016. The main limitation of the study was the unavailability of data on imaging information to adjudicate cases of hip fracture.

## Conclusions

Our results show that hip fracture incidence among patients aged 55 and over in China reached a plateau between 2012 and 2016. However, the absolute number of hip fractures and associated medical costs for hospitalization increased rapidly because of population aging.

### Author summary

#### Why was this study done?

- Hip fracture is a public health concern because of the associated considerable morbidity, excess mortality, great risk of disability, and heavy societal healthcare burden.

- Limited information on the incidence of hip fracture is available in low- or intermediate-income nations, especially on a national scale in China, despite China's having the world's largest population and rapid aging in recent years.

#### What did the researchers do and find?

- We conducted a retrospective cohort study using data on about 480 million residents from the national databases of Urban Employee Basic Medical Insurance and Urban Resident Basic Medical Insurance for 2012–2016 in China to examine age- and sex-specific incidence of hip fracture and associated costs for hospitalization.

- Our results show that incidence of hip fracture among patients aged 55 years and over in China was stable during the study period. However, the absolute number of hip fractures and associated total costs for hospitalization increased rapidly.

#### What do these findings mean?

- Incidence of hip fracture in China has stopped rising and has entered a plateau period.

- Further decline in hip fracture incidence is needed to reduce the absolute number of fractures and the socioeconomic burden resulting from population aging.

## Introduction

Hip fracture is the most devastating type of osteoporotic fracture because of the associated increased risk of morbidity [1], mortality [2–4], disability [5] and subsequent hip fracture [6], as well as high societal healthcare costs [7]. Studies have found that the 1-year mortality rate after hip fracture was as high as 22% [4] or even 30% [2]. Among those who survived, 50% lost functional independence, and one-third eventually became fully dependent [5]. With a rapidly aging global population, the hip-fracture-associated burden is expected to become a serious challenge.

It is projected that the number of hip fractures worldwide will increase from 1.26 million in 1990 to 4.5 million by 2050, about half of which are likely to occur in Asia, and particularly in China [1]. However, in the past decade, the incidence of hip fracture in high-income nations has declined or reached a plateau after an initial increase [8–13]. The vast majority of published studies have been conducted in high-income nations, and the data are scarce in low- or intermediate-income nations [14–16]. A systematic review conducted in 2014, including 22 studies from high-income nations/regions and 3 from low- or intermediate-income nations, indicated that hip fracture incidence began to decrease after an initial increase in high-income nations/regions, but still increased rapidly in low- or intermediate-income nations [17].

China, the largest low- or intermediate-income nation, has the largest population in the world and has been experiencing unprecedented rapid population aging. In 2018, the proportion of the population aged 65 years and older reached 11.93%, and this percentage is predicted to be 14% in 2025 and about 30% by 2050 [18]. Incidence of hip fracture and associated costs for hospitalization in China may place a significant burden on patients and their families, and are likely to be a challenge to Chinese basic medical insurance, with the increasing reimbursement ratio [19]. However, only a few studies have evaluated trends in hip fracture incidence and costs for hospitalization in China, using relatively small samples (830 [19], 938 [20], and 27,205 [21] patients), often over short time intervals (1 [20], 2 [21], and 3 [19] years) and covering only regional areas [16,22,23].

Accurate information about trends in the incidence of hip fracture and associated costs for hospitalization could ensure appropriate allocation of healthcare resources and effective implementation of policies to alleviate individual and social burdens. This study therefore aimed to evaluate trends in incidence of hip fracture and associated costs for hospitalization in China from 2012 to 2016.

## Methods

### Data sources

Data on hip fracture were obtained from the Urban Employee Basic Medical Insurance (UEBMI) and Urban Resident Basic Medical Insurance (URBMI) databases [24]. UEBMI covers working and retired employees in cities (i.e., employers and employees from government agencies and institutions, state-owned enterprises, private businesses, social organizations, and other private entities), and URBMI covers citizens without employment in cities (i.e., children, students, elderly people, and unemployed residents). By 2016, UEBMI and URBMI covered over 95% of the whole urban population of China. UEBMI and URBMI are updated on a monthly basis in all cities. Provinces/cities without detailed information on diagnosis were excluded. Data from about 480 million residents in 23 provinces (about 58.5% of the overall urban population in China) were included in this study. The variables included age, sex, International Classification of Diseases (ICD) code and name of primary and secondary diagnosis, and total medical expenses.

## Study population

Data were analyzed from a nationwide population covering approximately 100 million residents aged 55 years and older in 23 provinces over the 5-year period from January 1, 2012, to December 31, 2016. The time period of the study was chosen due to the availability of data. In patients aged 55 years and above, the most common cause of hip fracture is osteoporosis combined with low-energy trauma (generally arising from a fall) [25–28]. Hip fracture in younger patients is usually caused by high-energy trauma and is not related to osteoporosis [25–28]. In addition, indicators of osteoporosis such as bone density or imaging information are seldom available in the medical insurance databases. Therefore, consistent with previous publications on hip fracture incidence [8,9,11,16,29], in this study, hip-fracture-related indicators such as the incidence of hip fracture and associated costs in patients aged 55 years and above were used as surrogate indexes for determining the burden of osteoporosis or osteoporotic fractures. The inclusion criteria of the population were individuals who (1) had either UEMBI or URBMI during 2012 to 2016, (2) were enrolled in UEMBI or URBMI in 1 of the 23 provinces included in the study, and (3) were ≥55 years old at the first hip fracture during the study period. Individuals with the following conditions were excluded: (1) without a valid national insurance ID or (2) had conflicting information recorded, i.e., records of medical treatment reimbursement time earlier than the individual's first enrollment time. Insurance data were anonymized for study purposes. The study protocol was approved by the ethical review committee of the Peking University Health Science Center, and the requirement for informed consent was waived (IRB00001052-18012).

## Hip fracture identification

We identified hip fracture using ICD-9 code 820 and ICD-10 codes S72.0 and S72.1, following previous studies [9,15,30,31], and medical terms in Chinese including fracture of femoral neck, intertrochanteric fracture, and hip fracture. ICD-10 code S72.2 was excluded because of potential misclassification of femoral shaft fracture. To avoid missing patients when using the medical terms in Chinese, a fuzzy string matching algorithm was constructed to extract potential hip fracture patients from the database. Details of the method used are described in Section I of S1 Appendix. Briefly, the keywords were Chinese diagnostic terms written in different styles, including intracapsular fracture of the femoral neck, basicervical fracture of the femoral neck, subcapital femoral neck fracture, transcervical femoral neck fracture, and reverse obliquity intertrochanteric fracture. Additionally, we included specific symptoms and signs (hip pain, shorting and external rotation of the affected leg), special examination (hip X-ray), and particular treatments (open reduction and internal fixation of femoral neck fracture, open reduction and internal fixation of intertrochanteric fracture).

Exclusion criteria for case identification were the following: (1) pathological fracture, (2) old hip fracture, (3) femoral shaft fracture, (4) distal femoral fracture, (5) subtrochanteric fracture, (6) complications and sequelae of hip fracture (non-union, delayed union, malunion, osteomyelitis, osteoarthritis, and anchylosis), (7) prosthesis complications, (8) osteonecrosis of the femoral head, (9) hip dislocation, and (10) removal of internal fixation devices. In addition, patients with hip fracture have a 2-fold higher risk of second hip fracture than those without prior fracture [32]. Patients were consequently not included as a new case if (1) the fracture occurred within half a year after a previous one (this approach is consistent with previous publications [8]) or (2) the diagnosis text was clearly described as "old hip fracture" by doctors. Diagnostic information in the database of all hip fracture patients was reviewed independently by 2 orthopedic surgeons.

## Statistical analysis

National incidence for each of the 5 years from 2012 to 2016 was calculated. Incidence was stratified by sex and age (55–64, 65–74, 75–84, and ≥85 years old). Incidence was calculated using a 2-stage approach. Details of the method used to calculate incidence is provided in Section II of S1 Appendix. In brief, we first calculated age- and sex-specific incidence in each province. The denominator used to calculate the annual incidence of hip fracture was the total number of residents in UEBMI and URBMI in each province during the year. The numerator was the number of patients with hip fracture estimated for the denominator population in each province, taking into account missing data. The total enrolled population in each province can be separated into 3 groups: those with no records of a medical claim ($N_1$), those with complete information on diagnosis in claim records ($N_2$), and those with claim records missing diagnostic information ($N_3$). The observed incident patients with hip fracture ($M_1$) were from $N_2$, whereas the number of cases ($M_2$) in $N_3$ was estimated using a method based on Poisson regression with 10 rounds of multiple imputations. In the second stage, a random-effects meta-analysis was used to pool the province-specific estimates to calculate the national or regional (area) average estimates. We checked that the variance of province-specific estimates was stable using the Freeman–Tukey double arcsine transformation [33]. Incidence was expressed as per 100,000 person-years at risk, and 95% confidence intervals (CIs) were calculated assuming a Poisson distribution. Age-adjusted rates at the national level were calculated using China 2010 census data. Sensitivity analyses were conducted to assess the robustness of the results: (1) including only cases with complete data to assess the lower bounds of the rates and (2) excluding the top 10% of provinces ranked by the missingness of diagnostic information.

We also calculated hip-fracture-associated costs for hospitalization, including total costs and costs per patient. The costs in our study included only costs for inpatient hospitalization, which were categorized as diagnosis-related costs (including imaging investigations such as X-ray, computerized tomography, and magnetic resonance imaging), surgical treatment costs (associated with surgery, anesthesia, and medical materials), and pharmaceutical treatment costs (anti-osteoporosis drugs, etc.). Costs were discounted by the consumer price index (CPI) in each year to 2016 costs and converted into US dollars based on the 2016 RMB to US dollar exchange rate (period average). The CPI and exchange rate were from China Statistical Yearbook 2017.

Student's *t* test for continuous variables and the chi-squared test for categorical variables were used to compare statistics between males and females. The prespecified statistical analysis plan is provided in Section III of S1 Appendix. All statistical analyses used Stata version 15.0, and a 2-sided test with $P < 0.05$ was considered statistically significant. This study is reported as per the Strengthening the Reporting of Observational Studies in Epidemiology (STROBE) guideline (S1 STROBE Checklist).

## Results

Of all the 480 million residents in 23 provinces in the UEBMI and URBMI databases, a total of 102.56 million participants aged 55 years and older were included in this study (S1 Table), of whom 190,560 were confirmed as having a hip fracture during the study period. The locations of the 23 provinces are shown in Fig 1. Most hip fracture patients were Han Chinese (88.85%), 63.99% were female, and 42.88% of fractures occurred in those aged 75–84 years (Table 1).

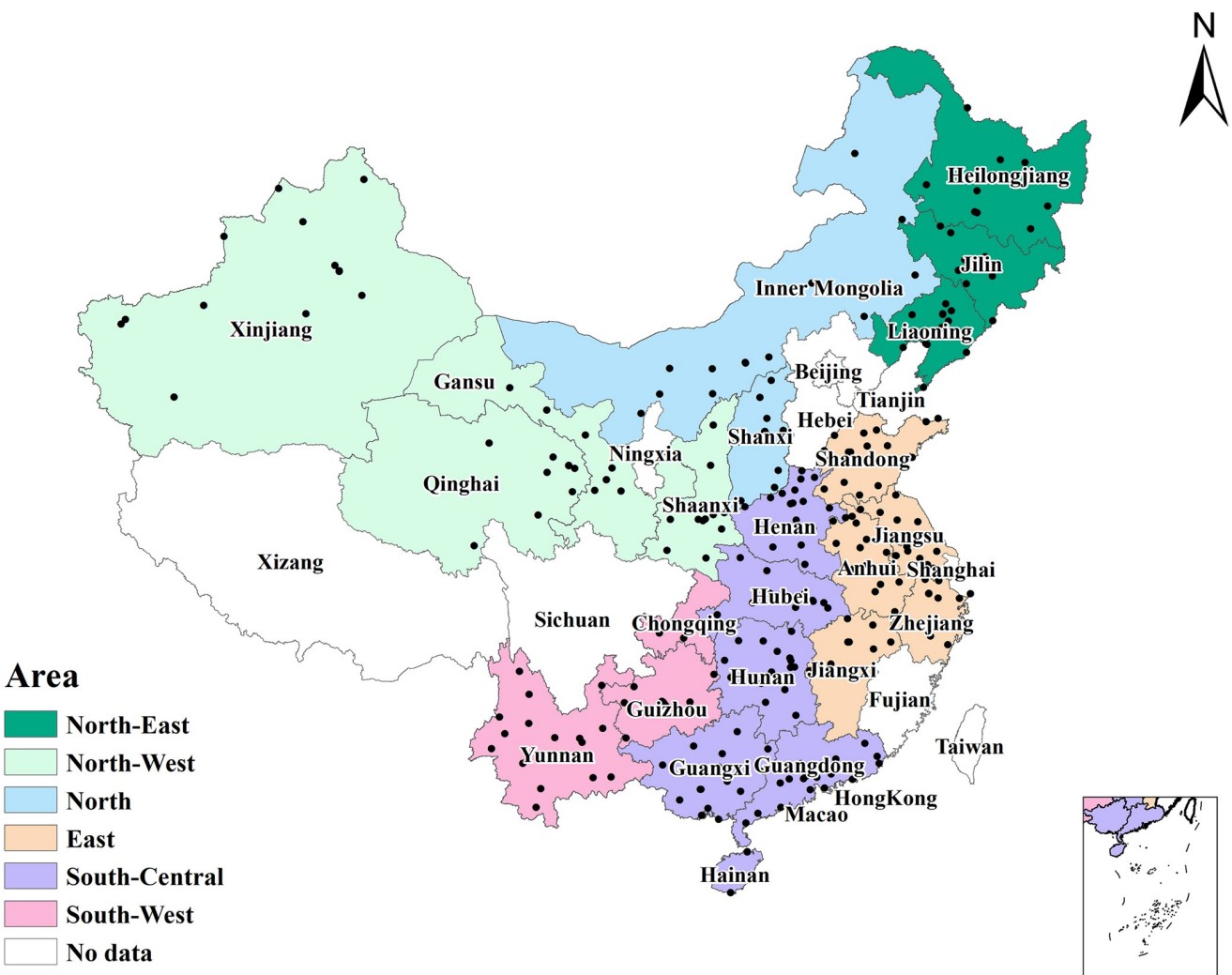

**Fig 1. The 23 provinces in China included in the study.** The geographical base map data were obtained from the Institute of Geographic Sciences and Natural Resources Research, Chinese Academy of Sciences (http://www.resdc.cn/). Black dots indicate the included cities in 23 provinces.

## Hip fracture incidence

Between 2012 and 2016, the total incidence of hip fracture in those aged 55 years and over was relatively stable, at 148.75 per 100,000 (95% CI 115.32–182.19; $P < 0.001$) in 2012 and 136.65 per 100,000 (95% CI 109.68–163.62; $P < 0.001$) in 2016. Similar patterns were also shown for both sexes. For females, the incidence was 180.72 per 100,000 (95% CI 137.16–224.28; $P < 0.001$) in 2012 and 177.13 per 100,000 (95% CI 139.93–214.33; $P < 0.001$) in 2016; for males, the incidence was 121.86 per 100,000 (95% CI 97.30–146.42; $P < 0.001$) in 2012 and 99.15 per 100,000 (95% CI 81.31–116.99; $P < 0.001$) in 2016 (Table 2; Fig 2A). When considering only cases with complete data, the lower bound of national incidence was 80.47 per 100,000 (95% CI 60.96–99.98; $P < 0.001$) in 2012 and 96.27 per 100,000 (95% CI 68.92–123.63; $P < 0.001$) in 2016, with a plateau (S2 Table). The result calculated by excluding the top 10% of provinces in terms of missing diagnostic information was also similar to the pattern reported above (S2 Table). Incidence increased with age, with the highest incidence in patients aged 85 years or older and the lowest incidence in those aged 55–64 years (Table 2; Fig 2B). The decrease was

**Table 1. Selected characteristics for hip fracture patients grouped by sex.**

| Characteristic | Subgroup | Total | Male | Female | Sex comparison statistics |
|---|---|---|---|---|---|
| Number | | 190,560 | 68,509 | 121,933 | |
| Age, years, mean (SD) | | 77.05 (8.94) | 76.12 (9.08) | 77.57 (8.82) | 32.855[a] |
| Age group, n (%) | 55–64 | 31,600 (16.58) | 13,222 (19.30) | 18,346 (15.05) | 912.884[b] |
| | 65–74 | 54,560 (28.63) | 20,397 (29.77) | 34,127 (27.99) | |
| | 75–84 | 81,706 (42.88) | 27,923 (40.76) | 53,749 (44.08) | |
| | ≥85 | 22,681 (11.90) | 6,963 (10.16) | 15,708 (12.88) | |
| Year, n (%) | 2012 | 16,587 (8.70) | 6,147 (8.97) | 10,419 (8.54) | 138.404[b] |
| | 2013 | 27,727 (14.55) | 9,970 (14.55) | 17,742 (14.55) | |
| | 2014 | 35,968 (18.87) | 13,760 (20.08) | 22,204 (18.21) | |
| | 2015 | 43,703 (22.93) | 15,526 (22.66) | 28,164 (23.10) | |
| | 2016 | 66,575 (34.94) | 23,106 (33.73) | 43,404 (35.60) | |
| Ethnicity, n (%) | Han | 169,315(88.85) | 61,317 (89.50) | 107,898 (88.49) | 81.683[b] |
| | Other | 17,827 (9.36) | 5,882 (8.59) | 11,935 (9.79) | |
| | Unknown | 3,418 (1.80) | 1,310 (1.91) | 2,100 (1.72) | |
| Area[c], n (%) | East | 76,485 (40.14) | 27,022 (39.44) | 49,426 (40.54) | 223.175[b] |
| | North | 4,381 (2.30) | 1,875 (2.74) | 2,500 (2.05) | |
| | North-East | 29,575 (15.52) | 10,321(15.07) | 19,253(15.79) | |
| | North-West | 4,284 (2.25) | 1,789 (2.61) | 2,492 (2.04) | |
| | South-Central | 51,392 (26.97) | 18,263 (26.66) | 33,110 (27.15) | |
| | South-West | 24,443 (12.83) | 9,239 (13.49) | 15,152 (12.43) | |

Thirteen patients (0.01%) had missing data for age; 118 patients (0.06%) had missing data for sex; 3,418 patients (1.79%) had missing data for ethnicity.

[a]Student's *t* test.

[b]Chi-squared test.

[c]East area included Jiangsu, Zhejiang, Anhui, Jiangxi, and Shandong provinces; North area included Shanxi and Inner Mongolia provinces; North-East area included Liaoning, Jilin, and Heilongjiang provinces; North-West area included Shaanxi, Gansu, Qinghai, and Xinjiang provinces; South-Central area included Henan, Hubei, Hunan, Guangdong, Guangxi, and Hainan provinces; South-West area included Chongqing, Guizhou, and Yunnan provinces.

SD, standard deviation.

more pronounced for patients aged 65 years or older in both sexes. However, the incidence remained broadly the same in those aged 55–64 years (Fig 2C and 2D).

## Absolute number of hip fractures

The overall number of hip fractures was 16,587, 27,727, 35,968, 43,703, and 66,575 for the years 2012, 2013, 2014, 2015, and 2016, respectively (Table 1; Fig 3A). The mean age at the first

**Table 2. Crude incidence of hip fracture grouped by sex and age group (per 100,000 person-years).**

| Characteristic | Subgroup | Hip fracture incidence (95% CI) per 100,000 person-years | | | | |
|---|---|---|---|---|---|---|
| | | 2012 | 2013 | 2014 | 2015 | 2016 |
| Total | | 148.75 (115.32–182.19) | 164.17 (133.84–194.50) | 138.98 (113.16–164.79) | 130.11 (105.87–154.34) | 136.65 (109.68–163.62) |
| Sex | Male | 121.86 (97.30–146.42) | 105.96 (84.38–127.54) | 108.17 (89.05–127.29) | 93.80 (76.59–111.00) | 99.15 (81.31–116.99) |
| | Female | 180.72 (137.16–224.28) | 204.04 (164.86–243.23) | 165.26 (132.63–197.89) | 156.48 (124.78–188.19) | 177.13 (139.93–214.33) |
| Age (years) | 55–64 | 46.30 (34.43–59.93) | 48.35 (37.80–60.18) | 40.32 (31.36–50.42) | 36.15 (27.35–46.17) | 38.48 (30.64–47.21) |
| | 65–74 | 131.50 (90.91–179.55) | 138.29 (105.14–175.97) | 113.81 (89.22–141.38) | 102.85 (79.93–128.66) | 115.16 (91.48–141.57) |
| | 75–84 | 342.80 (231.98–475.09) | 376.24 (280.45–486.02) | 319.98 (248.30–400.71) | 288.49 (224.37–360.63) | 314.99 (248.90–388.82) |
| | ≥85 | 548.87 (363.62–771.87) | 633.03 (448.60–848.91) | 568.47 (415.06–745.79) | 529.04 (395.52–681.83) | 523.96 (384.99–684.18) |

All rates are significantly larger than 0 with P < 0.001. The linear trend tests overall and for subgroup incidences across the 5 years were not significant (all P > 0.05).

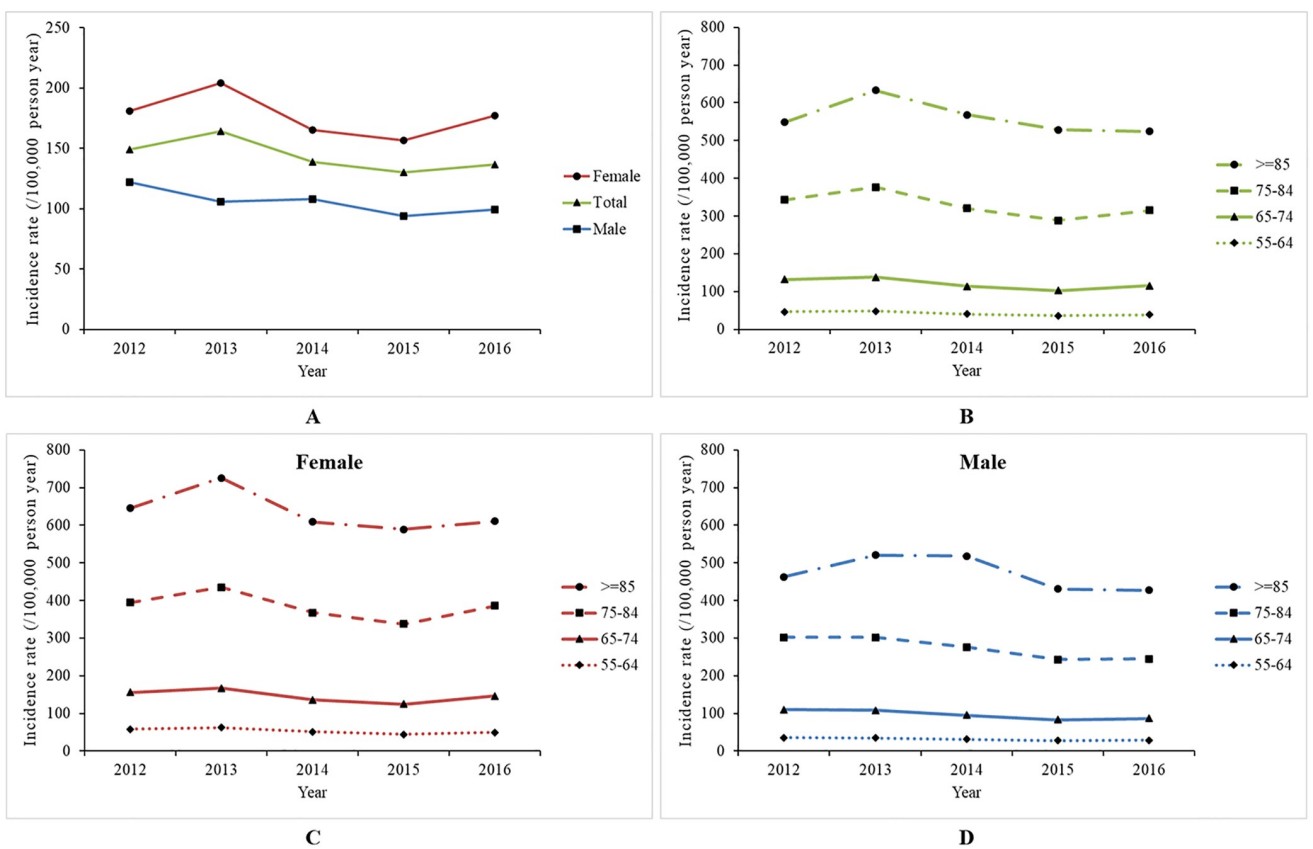

**Fig 2. Crude incidence of hip fracture in the population grouped by sex, age, and sex–age group (per 100,000 person-years).** (A) Hip fracture incidence by sex and year. (B) Age-specific hip fracture incidence. (C) Age-specific hip fracture incidence for females. (D) Age-specific hip fracture incidence for males.

hip fracture rose from 76.07 (SD 9.43) to 77.77 (SD 8.66) years over the 5-year period ($P < 0.001$; S1 Fig); the proportion of females was 62.89% in 2012, and increased to 65.26% in 2016 ($P < 0.001$).

## Standardization

When the incidence was standardized to the China 2010 census population, the total adjusted incidence of hip fracture was 128.10 (95% CI 88.68–174.79; $P < 0.001$) per 100,000 in 2012 and 114.46 (95% CI 89.85–142.06; $P < 0.001$) per 100,000 in 2016 (S3 Table).

## Costs for hospitalization for hip fractures

The total costs for hospitalization for hip fracture patients over the study period were US$1.049 billion, with an average of US$6,690 per patient. Total costs for hospitalization increased about 6-fold, from US$60 million in 2012 to US$380 million in 2016, while the costs per patient increased only 1.59-fold, from US$4,300 in 2012 to US$6,840 in 2016 (Fig 3B and 3C).

## Discussion

Using a nationally representative database covering approximately 480 million residents in 2012–2016, this study found that hip fracture incidence in China reached a plateau. In both

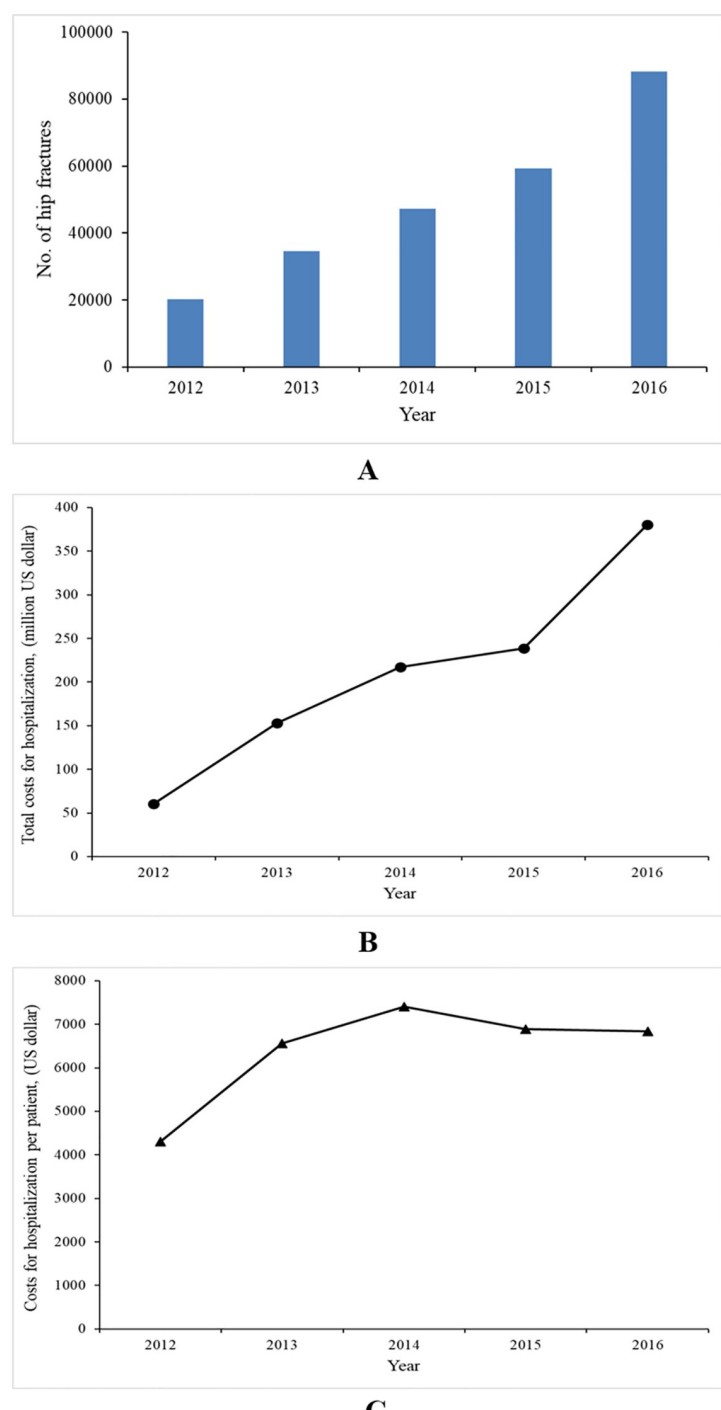

**Fig 3. Annual number of hip fracture patients and associated costs for hospitalization (in US dollars).** (A) Annual number of hip fractures. (B) Total costs for hospitalization. (C) Costs for hospitalization per patient. Costs were discounted by the consumer price index (CPI) in each year to 2016 costs and converted into US dollars based on the 2016 RMB to US dollar exchange rate (period average). The CPI and exchange rate were from China Statistical Yearbook 2017.

females and males, there was a decline in incidence in patients aged over 65 years, and incidence remained stable for those aged 55 to 64 years. However, the absolute number of hip fractures and associated costs for hospitalization increased rapidly.

The incidence of hip fracture has been shown to increase exponentially with advancing age [34], so it seemed likely that crude incidence would increase rapidly in China alongside the aging population. In our study, this plateauing incidence was very different from the previous rapid increase reported by Xia et al. in Beijing, China [16], or the pattern found in other low- or intermediate-income nations such as South Africa [14] and Kuwait [15]. However, most high-income nations, such as the US [8], Canada [9], Finland [10], France [11], Australia [12], and Sweden [13], have reported a plateau or continuing decline after an initial increase. A similar pattern of age-adjusted incidence lagging about 1 decade behind that of high-income countries was also found in Taiwan [35] and Hong Kong [36], 2 other economically advanced regions in China. The Hong Kong study additionally suggested that the decline may continue to occur a few years later in other Asian countries [36], and the speculation was recently confirmed in Singapore using national medical insurance claims data covering up to 5.6 million population [37]. In fact, Singapore and China were predicted to have a rapid rise in hip fracture incidence in the studies by Xia et al. and Koh et al. [16,38], but the latest data from both countries (Yong et al.'s study for Singapore [37] and our study for China) do not present that anticipated rising tendency. Additionally, a recent study reported that the hip fracture incidence decreased slightly in males and slowly increased in females in Tangshan city in China [22]. Hip fracture incidence in China between 2012 and 2016 may follow the same pattern as previously reported for most high-income nations.

However, the exact reasons for the plateau of hip fracture incidence in China have not been clearly explained. The rate of hip fracture among patients aged 55 years and above has been considered as the best estimate of osteoporosis or osteoporotic fractures [25,39]. Public health efforts regarding osteoporosis prevention and therapy in China might play an important role in this plateau of hip fracture incidence. First, 2 critical documents concerning prevention and treatment of osteoporosis, "Key Points on the Prevention and Treatment of Osteoporosis" [40] and "Guidelines for the Diagnosis and Treatment of Primary Osteoporosis" [41], were both issued in 2011 in China. Since then, osteoporosis has been listed as one of the National Health Priorities. The diagnosis rate of hip fracture and relevant preventive strategies were also subsequently improved after the release of the Chinese guidelines [42,43]. Similar policies and a decline of hip fracture incidence have also been seen in Taiwan of China since 2004 [44]. Second, zoledronic acid, available by prescription to postmenopausal women as the first-line treatment to prevent osteoporosis and reduce the risk of hip, vertebral, and other fractures [45], entered the Chinese market in 2009. Teriparatide, a novel bone anabolic agent, entered in 2011. Research showed that use of anti-osteoporotic drugs combined with calcium/vitamin D can reduce risk of re-fracture and mortality by 72.2% and 64%, respectively, compared with no treatment [46]. Studies in the US [8] and Hong Kong [36] suggested that the number of anti-osteoporosis treatments may be associated with the decline in hip fracture incidence, whereas a few other studies doubted this, as the decline appeared before the availability of bisphosphonate [9,47]. It is not clear and hard to determine how much the use of anti-osteoporotic medication accounts for the reduction of hip fracture incidence. However, the plateau occurred in China after 2012, which was close to the market release of zoledronic acid (2009) and teriparatide (2012). In addition, the rapid increase in use of anti-osteoporosis drugs was reported in different regions between 2008 and 2012 in China [48,49]. Compared with the hip fracture incidence during 2002–2006 in Beijing, China (229 per 100,000 person-years in females and 129 per 100,000 person-years in males) [16], the hip fracture incidence in our study was lower in both females and males, especially in 2016. Therefore, we speculate that the widespread

prescription of anti-osteoporotic drugs might have had an important impact on osteoporosis, and resulted in this plateau of hip fracture incidence. However, we acknowledge that, due to missing information on use of individual osteoporotic drugs, we cannot draw a direct causal relationship, and therefore further studies are still needed. Third, over 90% of hip fractures are attributed to a simple fall [50]. "Guidelines on Technical Interventions for Fall Prevention in Older People" was released by the Ministry of Health in China in 2011 [51]. Fall prevention strategies for secondary prevention of hip fracture have been identified as effective and have increased significantly in China, which may also have partly contributed to this plateau of hip fracture incidence [52,53].

In this study, the largest percentage decline in hip fracture incidence was in individuals aged 65 years and older in both males and females. In contrast, hip fracture incidence in those aged 55–64 years remained broadly constant from 2012 to 2016. These findings are consistent with findings in the US [8], France [11], and Taiwan of China [44], where older people showed a more pronounced decline. The underlying mechanism may be that older people benefit more from the prevention and treatment of osteoporosis because most hip fractures in this group are osteoporotic fractures caused by low-energy trauma [25].

Hip fracture incidence seemed to be stabilizing in China, but the number of hip fractures increased in all age groups in both males and females over the 5-year period. Our study found that the absolute number of hip fractures increased 4-fold. This was consistent with previous reports in Finland [54] and the US [55], where the number of hip fractures was also predicted to increase despite a decline in incidence. This apparent contradiction might reflect the serious aging problem in China. The mean age at first hip fracture increased in this study, which may also suggest the influence of an aging society.

The rapid increase of the absolute number of hip fractures meant that there was an approximate 6-fold increase in total annual costs for hospitalization for hip fracture from 2012 to 2016. Costs per patient also increased 1.59-fold. As mentioned above, Chinese guidelines and policies about osteoporosis prevention and therapy were issued in 2011, which encouraged doctors to prescribe medical treatments to these patients. In addition, costs of inpatient drugs containing anti-osteoporosis medication were reported to account for 25.2% of the total direct costs for osteoporotic fractures [20]. With the market release of zoledronic acid (2009) and teriparatide (2012), anti-osteoporosis drugs became more accessible. A rapid increase in the use of these drugs was also reported [48,49]. In addition, the increases in costs of surgery and associated medical materials might also contribute to the increase of the costs per patient. With a rapidly aging population, it is predicted that the percentage of the population aged 65 years and over will increase dramatically from about 11.93% in 2018 to 14% in 2025 and 30% by 2050 [18]. China is facing rapid population aging at a relatively early stage in its economic development. In addition, according to 2010 statistical data, there were only about 50,000 Chinese orthopedic surgeons, serving about 1.34 billion people [56]. Population aging and the increase in hip-fracture-associated costs for hospitalization are likely to become substantial challenges for clinicians and health policy-makers over the next generation, as they struggle to provide high-quality, cost-effective care to patients.

## Strengths and limitations

This study used a large, nationally representative sample of the Chinese mainland population, giving good estimates of the incidence and costs for hospitalization of hip fracture. It allowed us not only to provide an overall estimation of incidence rates but also to explore age and sex patterns in the rates across the country. This study also has several limitations. First, the database did not cover rural areas, which have a different insurance system. However, the

combined basic population structure was close to the distribution in the China 2010 population census data. Second, missing diagnostic variables could have affected the estimates. However, sensitivity analyses were used to explore the potential influence of these on the estimations. These included presenting the lower bound of the rates using only cases with complete data, to facilitate the interpretation of the findings. Finally, case ascertainment was limited, because the basic medical insurance databases did not provide information about laboratory data, imaging information, detailed surgical procedures, and cause of death. We were also unable to contact patients directly to obtain additional information because of the anonymity requirement.

## Conclusion

The incidence of hip fracture among patients aged 55 years and over in China reached a plateau between 2012 and 2016. However, the absolute number of hip fractures and associated medical costs for hospitalization increased rapidly because of population aging. Further decline in hip fracture incidence is needed to reduce the absolute number of fractures and the socioeconomic burden.

## Supporting information

**S1 Appendix. Statistical methods and analysis plan.**
(DOCX)

**S1 Fig. Change in mean age of hip fracture patients.**
(TIF)

**S1 STROBE Checklist. Checklist of items that should be included in reports of observational studies.**
(DOCX)

**S1 Table. Basic characteristics of the population aged 55 years and older in 23 provinces of China during 2012–2016.**
(DOCX)

**S2 Table. Sensitivity analysis of hip fracture incidence using only cases with complete data or excluding the top 10% of provinces ranked by the missingness of diagnostic information (per 100,000 person-years).**
(DOCX)

**S3 Table. Adjusted incidence of hip fractures (per 100,000 person-years).** Standardized by the population from the China 2010 census data.
(DOCX)

## Acknowledgments

We acknowledge Research Computing Platform from Peking University Health Science Center for assistance with data analysis and also thank Melissa Leffler, MBA, from Liwen Bianji, Edanz Editing China (www.liwenbianji.cn/ac), for editing the English text of a draft of this manuscript.

## Author Contributions

**Conceptualization:** Chenggui Zhang, Shengfeng Wang, Pei Gao, Siyan Zhan, Chunli Song.

**Data curation:** Chenggui Zhang, Jingnan Feng, Shengfeng Wang, Jinxi Wang.

**Formal analysis:** Jingnan Feng, Shengfeng Wang.

**Funding acquisition:** Siyan Zhan, Chunli Song.

**Investigation:** Chenggui Zhang, Jingnan Feng, Shengfeng Wang, Lu Xu, Lili Liu, Guozhen Liu, Jinxi Wang.

**Methodology:** Chenggui Zhang, Jingnan Feng, Shengfeng Wang, Pei Gao, Siyan Zhan, Chunli Song.

**Project administration:** Siyan Zhan, Chunli Song.

**Resources:** Siyan Zhan, Chunli Song.

**Software:** Jingnan Feng, Shengfeng Wang.

**Supervision:** Pei Gao, Siyan Zhan, Chunli Song.

**Validation:** Chenggui Zhang, Junxiong Zhu, Jialin Jia, Chunli Song.

**Visualization:** Chenggui Zhang, Jingnan Feng, Shengfeng Wang, Pei Gao.

**Writing – original draft:** Chenggui Zhang, Jingnan Feng, Shengfeng Wang.

**Writing – review & editing:** Chenggui Zhang, Jingnan Feng, Shengfeng Wang, Pei Gao, Lu Xu, Junxiong Zhu, Jialin Jia, Lili Liu, Guozhen Liu, Jinxi Wang, Siyan Zhan, Chunli Song.

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
