## [Editor Report · Decision Letter 0]

3 Feb 2020

Dear Dr Zhan, 

Thank you for submitting your manuscript entitled "Trends in Hip Fracture Incidence and Costs in China: A National Population Based Study, 2012-2016" for consideration by PLOS Medicine.

Your manuscript has now been evaluated by the PLOS Medicine editorial staff and I am writing to let you know that we would like to send your submission out for external peer review.

Kind regards,

Helen Howard, for Clare Stone PhD 

Acting Editor-in-Chief

PLOS Medicine 

plosmedicine.org

---

## [Decision Letter · Decision Letter 1]

30 Mar 2020

Dear Dr. Zhan,

Thank you very much for submitting your manuscript "Trends in Hip Fracture Incidence and Costs in China: A National Population Based Study, 2012-2016" (PMEDICINE-D-20-00290R1) for consideration at PLOS Medicine. 

[LINK]

In light of these reviews, I am afraid that we will not be able to accept the manuscript for publication in the journal in its current form, but we would like to consider a revised version that addresses the reviewers' and editors' comments. Obviously we cannot make any decision about publication until we have seen the revised manuscript and your response, and we plan to seek re-review by one or more of the reviewers. 

We expect to receive your revised manuscript by Apr 20 2020 11:59PM. Please email us (plosmedicine@plos.org) if you have any questions or concerns.

We look forward to receiving your revised manuscript. 

Sincerely,

Adya Misra, PhD

Senior Editor 

PLOS Medicine

plosmedicine.org

Please revise your title according to PLOS Medicine's style. Your title must be nondeclarative and not a question. It should begin with main concept if possible. "Effect of" should be used only if causality can be inferred, i.e., for an RCT. Please place the study design ("A randomized controlled trial," "A retrospective study," "A modelling study," etc.) in the subtitle (ie, after a colon).

The Data Availability Statement (DAS) requires revision. For each data source used in your study: 

Abstract

Last sentence of methods and findings should highlight a limitation of the study methodology/study design

Please provide brief demographics of the cohort

Conclusions- unclear how hip fractures over 55 are a surrogate for osteoporotic fractures. Please clarify and revise as needed 

* Please address the study implications without overreaching what can be concluded from the data; the phrase "In this study, we observed ..." may be useful.

* Please interpret the study based on the results presented in the abstract, emphasizing what is new without overstating your conclusions.

* Please avoid vague statements such as "these results have major implications for policy/clinical care". Mention only specific implications substantiated by the results.

* Please avoid assertions of primacy ("We report for the first time....")

Author summary

Please use an alternative for “most developed countries”. Do you mean specific countries or high-income nations? 

I’m not sure this can be directly inferred from your data so please revise “Our findings indicated that strategies for osteoporosis prevention and therapy may have been effective in China over the past few years”

Please rephrase “unparallel trends” to simpler and more accessible language

Introduction 

Since you cite previous studies on a similar topic, and note that they evaluated costs over a short time interval, please comment on why 3 years is considered a short time period but 4 (as in this case) isn’t? 

Please address past research and explain the need for and potential importance of your study. Indicate whether your study is novel and how you determined that. If there has been a systematic review of the evidence related to your study (or you have conducted one), please refer to and reference that review and indicate whether it supports the need for your study.

Methods

Please ensure that the study is reported according to the STROBE guideline. Please add the following statement, or similar, to the Methods: "This study is reported as per the Strengthening the Reporting of Observational Studies in Epidemiology (STROBE) guideline (SI file xxx)." 

Did your study have a prospective protocol or analysis plan? Please state this (either way) early in the Methods section.

Please clarify why the time period 2012-2016 was chosen? 

Please describe the inclusion and exclusion criteria in full- including age/co-morbidities etc

Please consider adding a map of the cities covered in this database? This might help with visibility and to demonstrate nationwide relevance of these data

Please clarify how you determined the age cut off of 55 years? 

Please describe the adjustments with China 2010 census data in the methods as noted in Fig 1. Same goes for the cost discounted using CPI, exchange rate etc in Fig 3

Please provide details of algorithm used and revise the term “relatively loose” to use more appropriate scientific language 

Please provide the keywords used (original language and English translation) as supplementary information such that the search may be replicated. 

STROBE checklist- please remove page numbers as these are likely to change. Instead you may use paragraphs and sections

Please revise “nearly decreasing trend” and use specific language instead 

Please provide p values along with 95% confidence intervals. Please note exact p values should be provided if p>0.05 or p<0.05, p<0.001 is permitted. 

Discussion

Please present and organize the Discussion as follows: a short, clear summary of the article's findings; what the study adds to existing research and where and why the results may differ from previous research; strengths and limitations of the study; implications and next steps for research, clinical practice, and/or public policy; one-paragraph conclusion.

Please clarify if zoledronic acid is provided to those at risk of osteoporosis or is available over the counter etc

Comments from the reviewers:

Reviewer #1: Comments to Authors

Overall, this is a well-written and interesting manuscript describing the incidence and costs of hip fractures in China.

There are only a few minor points that could be clarified prior to publication:

1. The English language is very good, there are just a few minor grammatical issues which can be edited during the production process.

2. The sentence "Further decline is needed to reduce the absolute number of fractures and socioeconomic burden resulting from population aging" is a clear conclusion. It would be better if the similar sentences in the Abstract and Conclusion could be replaced by this sentence instead. i.e. use this sentence instead of "Further decline is needed to reduce the heavy burden imposed by hip fracture".

3. In the Introduction section, it would be useful to provide a reference for the sentence "It is projected that the number of hip fractures worldwide…"

4. It would be useful to move the sentence "To reduce double-counting, subsequent fractures were considered…" to the section titled "Hip Fracture Identification."

5. In the Results section, it could be useful to include a paragraph describing the absolute number of fractures.

6. In the Discussion section, could the authors discuss the reasons that the costs per patient for hip fracture have increased? The authors have provided an excellent discussion regarding the number and rates of hip fractures, if they could also discuss costs per patient, that would strengthen the Discussion further.

7. References 15 and 16 do not seem to be appropriate. It seems that these references were used because the authors used similar methods. It would be useful if the authors could find the original reference for that method. Otherwise, they should specify in the text that they completed their analyses using methods similar to these other studies.

8. Reference 29 does not seem to be properly listed. Could the authors please check this?

9. In Table 1 and Figure S1, the authors report both median and mean ages. If the data is normally distributed, the authors should report age as mean±SD. If not, then median(IQR). Could the authors please check their data and use either mean or median, but not both?

10. A map of the region would be useful, instead of a list of provinces (Table S1). Could the authors please provide a map, which will help international readers understand the region better?

11. Could the authors please provide a more descriptive caption for Table S3?

Reviewer #2: I confine my remarks to statistical aspects of this paper. These were very well done, but I have a couple of minor points to address before I can recommend publication.

Overall, I would de-emphasize discussion of trends. The data in the table and figures don't show a linear trend and it's hard to see what trend they might show. It looks like noise.

In Table 1, I would delete the p values - with N this large, everything is significant. And please list what staistic (t, chi square or whatever) is in the "Statistics" column

Figures 1 and 3 - I don't really like graphs with two axes - changing the limits of either axis changes the appearance a lot. Decide on what you want to show. If it's the relation between total cases and incidence/cost then a ratio could be plotted, or you can make a scatter plot of the two, with directed arrows showing trend over time. If that is not what you want to show, then either show only one set of data or adapt something to sut what you want to show.

Peter Flom

Reviewer #3: The methods are too brief to allow the reader to understand the results and require more detail. 

1. Case ascertainment is very important. More information is needed for the natural language processing in terms of training and test, internal / external validation. Were there changes in the quality of the text records over time. 

2. What proportion of patients had surgery, did this change over time. Usually validation of cases requires surgical / procedure codes. Were the types of patients identified by procedure codes different to those without. 

3. What is the definition of old hip fracture. 

4. How were cases really validated. It seems unfeasible for two surgeons to validate 190,560 cases. 

5. The numerator is described as taking into account missing data. What does this mean? Imputation? 

6. The incidence seems to be per province population. I assume there was standardisation for both age and sex. 

7. Is the reason for the decrease in incidence per population due to differences in age demographic over time. 

8. It is not clear how costs were calculated and over what time period. This needs to be added to the methods

9. What was the one year mortality, did it vary over time, sex and age group. 

10. Reductions in age specific hip fracture rates have been seen in other populations, often preceding DXA and pharmacological uptake in the community. Please comment

[LINK]

---

## [Decision Letter · Decision Letter 2]

20 May 2020

Dear Dr. Zhan,

Thank you very much for re-submitting your manuscript "Incidence of and trends in hip fracture among adults in urban China: a nationwide retrospective cohort study" (PMEDICINE-D-20-00290R2) for review by PLOS Medicine.

I have discussed the paper with my colleagues and the academic editor and it was also seen again by xxx reviewers. I am pleased to say that provided the remaining editorial and production issues are dealt with we are planning to accept the paper for publication in the journal.

[LINK]

We look forward to receiving the revised manuscript by May 27 2020 11:59PM. 

Sincerely,

Adya Misra, PhD

Senior Editor 

PLOS Medicine

plosmedicine.org

Requests from Editors:

Abstract conclusions

Please start this section with “ our results show..” or similar

In the abstract, please clarify if the numbers in brackets represent ranges or 95% confidence intervals? It would be god to clarify this within the entire text as well, where you may have provided one or the other. Please also provide p values, where you provide 95% Confidence intervals

Please use the "Vancouver" style for reference formatting, and see our website for other reference guidelines https://journals.plos.org/plosmedicine/s/submission-guidelines#loc-references

Please add a single sentence referring to Figure 1 within the main text. Could you confirm that this is an original image and there are no copyright restrictions on sharing this map?

Please state whether any changes were made to the prespecified analyses, within the methods section

Please ensure the reference brackets are placed after a space in text and then followed by full stop or comma as needed. For example xxx [1,2]. 

Comments from Reviewers:

Reviewer #1: The authors have done an excellent job of responding to all the reviewer and editor comments. I believe that this manuscript is now acceptable for publication.

Reviewer #2: The authors have addressed my concerns and I now recoomend publication.

Peter Flom

[LINK]

---

## [Editor Report · Decision Letter 3]

13 Jul 2020

Dear Prof. Zhan, 

On behalf of my colleagues and the academic editor, Dr. Kassim Javaid, I am delighted to inform you that your manuscript entitled "Incidence of and trends in hip fracture among adults in urban China: a nationwide retrospective cohort study" (PMEDICINE-D-20-00290R3) has been accepted for publication in PLOS Medicine. 

PRODUCTION PROCESS

PRESS

PROFILE INFORMATION

Thank you again for submitting the manuscript to PLOS Medicine. We look forward to publishing it. 

Best wishes, 

Adya Misra, PhD

Senior Editor 

PLOS Medicine

plosmedicine.org